# Effects of Pre-Existing Mental Conditions on Fatigue and Psychological Symptoms Post-COVID-19

**DOI:** 10.3390/ijerph19169924

**Published:** 2022-08-11

**Authors:** Stefanie Homann, Marie Mikuteit, Jacqueline Niewolik, Georg M. N. Behrens, Andrea Stölting, Frank Müller, Dominik Schröder, Stephanie Heinemann, Christina Müllenmeister, Iman El-Sayed, Christine Happle, Sandra Steffens, Alexandra Dopfer-Jablonka

**Affiliations:** 1Department of Rheumatology and Immunology, Hannover Medical School, 30625 Hannover, Germany; 2Dean’s Office, Hannover Medical School, 30625 Hannover, Germany; 3Department of General Practice, University Medical Center Göttingen, 37073 Göttingen, Germany; 4Department of Pediatric Pneumology, Allergology, and Neonatology, Hannover Medical School, 30625 Hannover, Germany; 5German Center for Lung Research, Biomedical Research in End Stage and Obstructive Lung Disease—BREATH, 30625 Hannover, Germany; 6Excellence Cluster for Infection Research RESIST—Resolving Infection Susceptibility, 30625 Hannover, Germany; 7German Center for Infection Research (DZIF), Partner Site Hannover-Braunschweig, 30625 Hannover, Germany

**Keywords:** post-COVID-19, fatigue, anxiety, depression, psychological pre-conditions

## Abstract

Background: Patients who are post-COVID-19 will require more treatment soon. Therefore, it is important to understand the root cause of their psychological and somatic conditions. Previous studies showed contradictory results on the influence of pre-existing mental conditions. The present study examines the influence of these pre-existing conditions and their pre-treatment on the severity of post-COVID-19 symptoms. Methods: This analysis employs questionnaire data from a large study sample in Germany. Overall, 801 participants were included. All participants rated their health status on a scale from 0 to 100. Fatigue, depression, and anxiety were measured using the FAS, PHQ-9, and GAD-7 scales. Results: All pre-pandemic values showed no significant differences between the groups. The current health status was rated similarly by the recovered patients (μ = 80.5 ± 17.0) and the control group (μ = 81.2 ± 18.0) but significantly worse by acutely infected (μ = 59.0 ± 21.5) and post-COVID-19 patients (μ = 54.2 ± 21.1). Fatigue, depression, and anxiety were similar for recovered patients and the control group. By contrast, there were significant differences between the control and the post-COVID-19 groups concerning fatigue (45.9% vs. 93.1%), depression (19.3% vs. 53.8%), and anxiety (19.3% vs. 22.3%). Conclusion: Fatigue and psychological conditions of post-COVID-19 patients are not associated with pre-existing conditions.

## 1. Introduction

The coronavirus SARS-CoV-2 (severe acute respiratory syndrome coronavirus type 2) has caused a global pandemic, with more than 539 million confirmed infections and more than 6 million deaths as of June 2022 [1]. It is assumed, that 10% to 35% of people are expected to have persisting and long-lasting symptoms following acute COVID-19 disease [2]. In hospitalized COVID-19 patients, an even higher prevalence of about 50% cases with long-lasting symptoms was reported [3]. This condition is often referred to as “post-acute COVID-19”, “long haulers”, “long-COVID”, and others. The World Health Organization (WHO) recently introduced the “post-COVID-19” condition as a symptomatic condition after an acute SARS-CoV-2 infection, three months from the onset of acute symptoms that last at least for two months and cannot be explained by an alternative diagnosis [4]. Post-COVID-19 affects multiple organs and is associated with a broad spectrum of possible symptoms, including muscle pain, shortness of breath as well as cardiovascular, cognitive, neurological, and psychological impairments [5]. Similarly, long-term symptom persistence is also prevalent among other viral infections. For instance, 40% of SARS-CoV-1 patients suffered from chronic fatigue four years after the disease [6]. Immunological processes causing a persistent inflammation have been supposed as a likely source of post-COVID-19 [7]. While acute COVID-19 has been extensively investigated, in-depth studies of the health consequences of post-COVID-19 are still pending [8].

Fatigue has been identified as one key symptom in post-COVID-19 and has been attributed to impairment of multiple organs (including the lungs, the heart, and the peripheral nervous system), and inflammatory processes. Factors contributing to fatigue can be grouped as (1) factors affecting the central nervous system (e.g., neurotransmitter levels, inflammation, etc.) (2) psychological factors (e.g., fear, anxiety, stress, and depression), and (3) peripheral factors (muscular susceptibility to the virus and others) [6]. The outlined psychological factors especially are widely spread in Western countries.

The 12-month prevalence of anxiety was reported as 14.0% for Europe [9] and 15.3% in Germany [10]. The 12-month prevalence of depression was reported as 6.9% [9] and 8.1% [11] for Europe and Germany, respectively. There is a growing body of evidence that an acute COVID-19 infection can worsen depression and anxiety symptoms [12,13,14].

A recent systematic review by Schou et al. [15] confirms fatigue, anxiety, and depression among the most common sequelae of COVID-19. Anyhow, Magnusdottir et al. [16] stress that little is known about the long-term mental health of COVID-19 patients. In their observational study across six nations, they find that the severity of the acute disease plays a crucial role: patients with a light course of disease exhibited lower risk of depression and anxiety relative to the reference group, while the opposite was found for patients bedridden for more than seven days.

Pre-existing mental disorders are reported to increase the risk of COVID-19 severity and mortality [17]. There is some agreement that a previous psychiatric history also is a predictor of depression and anxiety [17,18,19,20] following an acute COVID-19 infection. Furthermore, it was hypothesized that pre-existing psychological and psychosomatic conditions are associated with the occurrence and worsening of psychological post-COVID-19 symptoms and fatigue [6,21].

This study investigates if physical and psychological disorders in post-COVID-19 patients can be attributed to psychological pre-conditions. To this end, we surveyed for fatigue and the predominant psychological symptoms (anxiety, affective disorders/depression, somatic pain disorders) as well as psychological or psychotherapeutic treatment prior to a COVID-19 infection.

## 2. Materials and Methods

### 2.1. Study Design and Participants

The study is based on the online based longitudinal observational study DEFEAT Corona (DEFEnse Against COVID-19-STudy-Looking forward) conducted in the German Federal State of Lower Saxony (approx. 9 million inhabitants). As only data from one survey was used for this analysis, this current study can be considered a cross sectional survey. Study participants could enroll in the study by filling out the online survey on our website www.defeat-corona.de (accessed on 7 July 2022). Before enrollment, participants were asked for consent to participate in the study and needed to confirm that they were 18 years or older. Anyone was eligible to enroll, regardless of whether a participant had COVID-19 or not, or if a participant had experienced long-lasting symptoms.

The SoSci Survey online platform (SoSci Survey GmbH, Munich, Germany) was used to create the questionnaire. The questionnaire was only available in the German language.

Potential participants were informed about the study by posters including QR-codes linking to the study website. These posters were distributed in Fall 2021 in public locations in Lower Saxony, e.g., town halls, churches, or public libraries as well as in medical facilities. Additionally, we asked 400 randomly selected doctors working in primary care to display fliers in the waiting rooms of their practices. To specifically reach people having post-COVID-19, we asked selected post-COVID-19 patient advocacy groups to share the link to our survey on their social media channels.

Due to our extensive set of items, the survey was split into two parts requiring about 30 min each. Participants could always interrupt to fill out the questionnaire and resume it at another time. Data collection of the first part of the survey was started in September 2021. Some days after completion, each participant was asked to take part in the second part. This study included survey responses up to the beginning of March 2022.

According to their COVID-19 disease history, four groups were formed and compared for the present analysis: (1) Participants not infected by SARS-CoV-2 (in the following referred to as the “reference group”), (2) participants with a previous COVID-19 infection who were fully recovered (in the following referred to as the “recovered group”), (3) participants with an acute SARS-CoV-2 infection or previous infection within the last 90 days and ongoing pro-longed symptoms (in the following referred to as the “prolonged COVID” group), (4) participants with a previous COVID-19 infection and persisting symptoms for more than 90 days (in the following referred to as the “post-COVID-19” group). Participants who had not answered the second questionnaire before the beginning of March 2022 or who stated that their COVID-19 diagnosis was neither confirmed by a PCR nor an antigen test were excluded from the evaluation.

### 2.2. Measurements

We collected various self-reported socio-demographic and health-related information. Demographic data included gender, age, family status, level of education, occupation status, and history of migration. Health-related information included the participants’ current physical and mental health status as well as the respective values before the pandemic.

#### Health Status before the Pandemic

To assess health prior to the pandemic, participants were asked to retrospectively rate their health status. The self-assessed health status was obtained using a visual analog scale (VAS) of the EQ-5D-VAS [22] ranging from 0 to 100 indicating very poor health (0) to excellent health (100). Additionally, the subjects were asked to checkmark if they had suffered from depression, anxiety disorders, chronic exhaustion, or burnout prior to the pandemic. We further inquired whether participants had ever undergone prior psychotherapeutic or psychiatric treatment or received drug therapy for a mental health condition (incl. antidepressants, sedatives, or anti-psychotic drugs).

#### Current Health

We used the EQ-5D-VAS again to assess the participants’ current general health status. Fatigue was measured using the German version of the Fatigue Assessment Scale (FAS), a 10-item scale evaluating symptoms of chronic fatigue including both mental and physical symptoms with five items each [23,24]. The FAS showed good internal consistency (α = 0.90) and reliability [25]. Items are to be answered using a five-point Likert-type scale ranging from 1 (“never”) to 5 (“always”). Sum scores can range from 10 (indicating “no fatigue”) to 50 (denoting “highest fatigue”). We used predefined score ranges, where a total score of 10 to 21 indicates no fatigue and scores 22 to 50 indicates substantial fatigue [26].

The German version of the Patient Health Questionnaire (PHQ-9-D) was used to screen for presence and severity of depression. This self-reported scale consists of nine items that can be answered on a four-point scale and was developed to diagnose presence and severity of depressive symptoms in primary care settings [27,28]. A PHQ-9 sum score ≥10 is a typical threshold to indicate major depressive symptoms (sensitivity 88%, specificity also 88%).

The German version of the Generalized Anxiety Disorder Scale (GAD-7) was used to determine the participants’ grade of anxiety. The scale consists of seven items that can be answered on a four-point scale and showed a high internal consistency (α = 0.89). A GAD-7 sum score value ≥10 is considered as presence of clinically relevant anxiety symptoms [28]. Sensitivity and specificity for this score threshold were 89% and 82%, respectively [29].

#### Statistical Methods

Statistical analysis was carried out as a descriptive/explorative analysis using SPSS 27 (IBM corp., Armonk, NY, USA). First, we assessed metric variables whether the data was normally distributed using the Kolmogorov–Smirnov test. In case of a normal distribution, the student’s *t*-test was used to compare two and the ANOVA test to compare more than two groups. If values were not normally distributed, the Mann–Whitney-U test and the Kruskal–Wallis test were used to compare two or more groups, respectively.

When comparing nominal items between two groups Pearson’s chi-square test was employed, and Cramer’s V was used when comparing more than two groups. When a sample group contained less than five participants, Fisher’s exact test was applied. *p*-values < 0.05 are considered significant.

#### Research Ethics

The study received approval from the responsible ethics boards of Hannover Medical School (9948_BO_K_2021) and University Medical Center Göttingen (29/3/21). The study is registered in the German register for clinical trials (DRKS00026007).

## 3. Results

Out of 1258 initial study participants, 801 participants met inclusion criteria and were included in the analysis (cf. Figure 1).

### 3.1. Demographics

Table 1 compares all four groups with respect to gender, age, and family status. No significant differences in age and family status were found. Only the gender variable showed differences between the groups: there were significantly more women (81.4%) in the post-COVID-19 group than in the other three groups (reference 67.9%, recovered 77.0%, prolonged COVID-19 69.0%).

### 3.2. Health Status Prior to the COVID-19 Pandemic

Regarding the subjective rating of the overall health status prior to the pandemic, there were no significant differences between any of the four groups. Average values on the subjective 100-value scale were (with standard deviations in brackets): reference group μ = 80.1 (± 17.8), recovered patients μ = 84.8 (± 11.1), prolonged COVID group μ = 82.3 (± 15.6), and post-COVID patients μ = 82.4 (± 15.0).

Results for pre-existing psychological conditions of the different groups are compared in Table 2. There were no significant differences in the occurrence of almost all of the investigated disorders (depressions, anxiety disorder, chronic pain, and burnout). Likewise, we found neither significant differences in psychotherapeutic or psychiatric treatment nor in medication with psychiatric drugs prior to the pandemic. As the only exemption, post-COVID-19 patients reported significantly lower fatigue prior to pandemic (*p* = 0.02).

### 3.3. Health Status after COVID-19 Infection

Whereas the groups reported similar health status prior to the pandemic, the comparison of their current health figures exhibits significant differences. This is visualized in Figure 2. The reference group rated their health equally good μ = 81.2 (± 18.0) prior to the pandemic. This is also true for the recovered group μ = 80.5 (± 17.0). By contrast, the prolonged and post-COVID-19 groups reported a highly significant reduction to μ = 59.0 (± 21.5) and μ = 54.2 (± 21.1), respectively.

When comparing the means of the reference and recovered groups against each other, no significant differences were found. The same holds when comparing the prolonged and post-COVID-19 groups against each other. However, significant differences between the reference and recovered groups to the prolonged and post-COVID-19 groups were found.

### 3.4. Psychological Symptoms after the COVID-19 Infection

The current psychological state was compared in terms of fatigue, depression and anxiety as visualized in Figure 3. The evaluation of the fatigue assessment scale (FAS) showed very high differences between the groups. Fatigue (FAS ≥ 22) was observed in 45.9% and 42.0% of the reference and recovered groups, respectively. These two did not differ significantly. This already unexpectedly high level was further markedly increased to 84.1% and 93.1% for the prolonged COVID-19 and post-COVID-19 groups. Fatigue in the post-COVID-19 group was significantly higher than in the prolonged COVID-19 group. Both these groups showed highly significantly increased fatigue values when compared to the former two groups.

The evaluation of the depression questionnaire PHQ-9 showed that 23.6% of the reference group had depression symptoms, which is significantly higher than the 13.3% of recovered patients. By contrast, 54.8% and 54.6% of the prolonged and post-COVID-19 patients in our survey exhibited depression. Both being significantly above the former two groups. Prolonged and post-COVID-19 groups did not differ significantly from each other in terms of depression.

The participants’ worriedness was measured on the generalized anxiety disorder assessment scale (GAD-7). Notably, only 7.2% of the recovered patients showed anxiety on this scale which is significantly less than the reference group (15.8%). By contrast, the prolonged COVID-19 group (26.2%) and post-COVID-19 patients (22.7%) were affected significantly more often. In the post-COVID-19 group anxiety was also significantly higher than in the reference group.

## 4. Discussion

We conducted a cross-sectional study including 801 persons. Post-COVID-19 patients formed the largest group (n = 395). The reference group was only slightly smaller (n = 274). The groups of recovered and prolonged COVID-19 patients were much smaller, as this was not the main focus of the cohort.

Our first key finding is that all groups did not differ significantly concerning their mental health conditions prior to the pandemic. This holds for the self-assessment of the overall health status as well as for psychological disorders (including depression, anxiety, burnout, chronic pain, exhaustion, and other factors) and psychotherapeutic or psychiatric treatment. Prior fatigue was even reported lower by the post-COVID-19 patients.

This could have two implications: First, a preexisting psychological condition does not impose an increased vulnerability to developing a post-COVID-19 syndrome in our cohort. Second, this indicates that the subsequently reported worsening in the health status was mostly developed during the COVID-19 infection and cannot be attributed to preexisting and diagnosed psychological disorders.

The recovered participants show no worse physical condition than the reference group in terms of subjective health status and fatigue. The mental symptoms depression and anxiety were slightly, but significantly, reduced. This is in agreement with previous observations by Magnusdottir et al. [16]. Notably, the depression and anxiety values reported by recovered patients are similar to the baseline values published prior to the pandemic [9,10]. This might be explained by personal relief after getting through the disease and increased confidence in their body’s ability to ward off infection.

The second key finding of this study is that fatigue, depression and anxiety are significantly increased in prolonged and post-COVID-19 despite comparable pre-existing psychological conditions as the reference group. This means that these symptoms have likely developed because of the COVID-19 infection and the accompanying physical and social stress and should be treated as such.

Comparability of our absolute numbers to published values is limited, since the surveys employed in this study focused on the individual assessment of physical and mental status before the disease and the current conditions. Depression values (23.6%) of our reference group were higher than previously reported prevalence values. This observation may be partially due to the general pandemic situation and is less pronounced in the group of recovered patients. Anxiety values reported by our reference and recovered groups are within the span of published values [9].

Overall, the present study supports the understanding that the increased and persisting psychological symptoms in this study’s participants are mostly a consequence of their COVID-19 infection rather than a continuation of a pre-existing condition, as also suggested by Koczulla et al. [30]. To prevent a mutual intensification and chronification of physical and mental symptoms, an early diagnostic and therapeutic approach is needed to reduce disease severity and mortality [31].

As no satisfactory therapy for fatigue is known to date, a patient’s individual coping (including motivation, sleeping habits, and other factors) with this syndrome plays a major role [32]. A psychotherapeutic treatment in addition to pharmacological treatment can also alleviate symptoms and avoid chronification.

Any intervention should ultimately aim to strengthen personal resources, reduce stress, and enable adequate coping behavior [30]. Behavioral changes and psycho-social measures can help to reduce stress, and consequently also have a positive effect on immune responses in viral respiratory infections [33].

Limitations of the study include: Regarding demographic factors, our study involved more women than men among all participants. Especially, in the post-COVID-19 group more than four-fifths were female. This relation has already been observed by Bai et al. [34]. Furthermore, the health status prior to the pandemic could only be recorded retrospectively and the questionnaires we only provided in the German language. Despite these limitations, our dataset is valid for a comparative study. Notably, our groups showed no significant differences with respect to age and family status.

## 5. Conclusions

Our post-COVID-19 patients report significantly increased fatigue, depression and anxiety while the pre-pandemic physical and mental condition was comparable to the reference group. This means that these post-COVID-19 symptoms cannot directly be attributed to their pre-conditions but have rather developed over the course of the disease.

Participants who recovered from COVID-19 rated their health status as well as fatigue, depression, and anxiety even better than the reference group. This indicates that convalescing from the disease not only means complete physical and mental restoration, but may even improve personal confidence and coping.

By contrast, persons with prolonged COVID-19 or post-COVID-19 assessed their health status significantly worse and reported fatigue and depression twice as often as well as slightly increased anxiety. Notably, the absolute portion of prolonged and post-COVID-19 patients with fatigue (>80%) and depression (>50%) is very high.

In conclusion, this present study underlines that fatigue and psychological symptoms should be mostly considered as a consequence of the COVID-19 infection. Fatigue and depression were simultaneously increased and persisting. It can be assumed that these two factors mutually aggravate each other, as also suggested by Rudroff et al. [6], due to reduced physical activities and social isolation during the pandemic.

To overcome this negative feedback circle, timely treatment of the patients is required. As fatigue is difficult to treat directly, a psychotherapeutic approach may help to address the specific situation of post-COVID-19 patients. Rather than on the biographical background, the therapy should focus on personal experiences since the COVID-19 infection and coping with the persisting symptoms.

Especially a supportive-expressive group therapy is known to be an effective treatment of patients with inflammatory and autoimmune diseases [35]. This can effectively enable patients to increase their physical activities, reduce stress and anxiety as well as overcome social isolation and inner loneliness. A group therapy represents a suitable treatment offer that is needed to treat the large numbers of patients affected by post-COVID-19.

## Figures and Tables

**Figure 1 ijerph-19-09924-f001:**
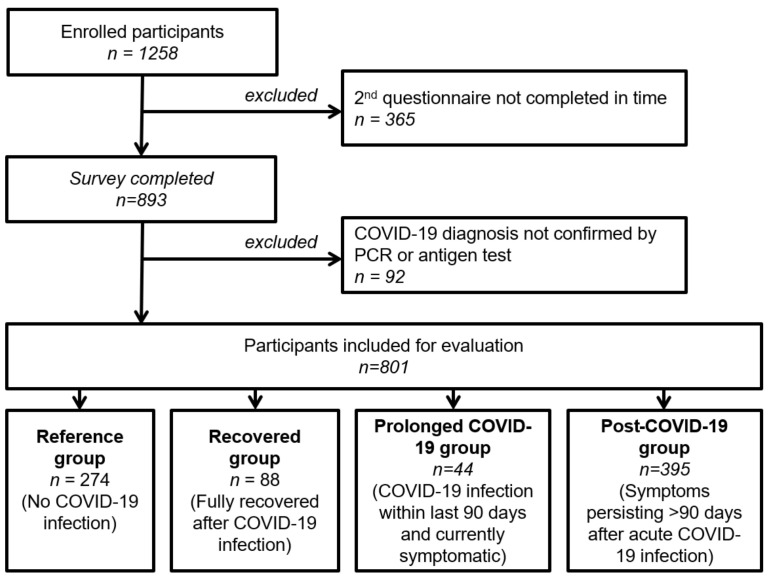
Flowchart of patient inclusion.

**Figure 2 ijerph-19-09924-f002:**
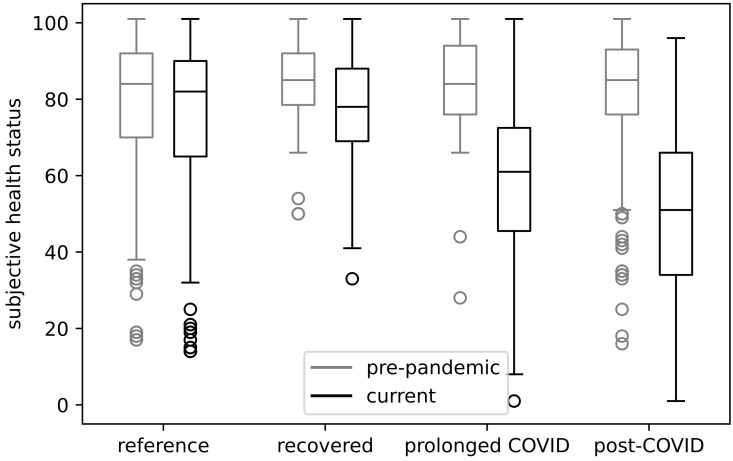
Assessment of health status before the pandemic and now.

**Figure 3 ijerph-19-09924-f003:**
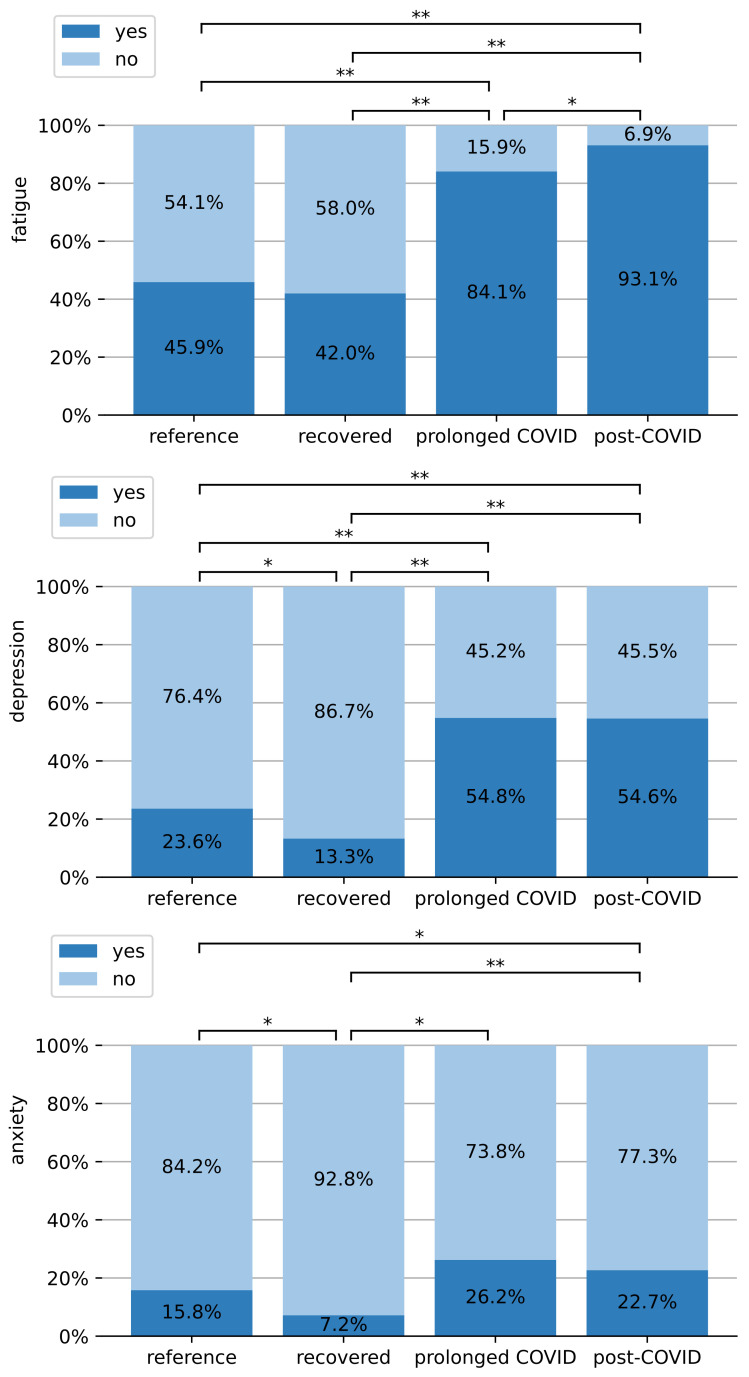
Comparison of fatigue (FAS ≥ 22), anxiety disorders (GAD-7 ≥ 10) and depression (PHQ-9 ≥ 10) between all groups. Symbols: *: *p* ≤ 0.05, **: *p* ≤ 0.001.

**Table 1 ijerph-19-09924-t001:** Demographic data.

	Reference Group (n = 274)		Fully Recovered (n = 88)		Prolonged COVID-19 (Symptoms for ≤ 90 Days) (n = 44)		Post-COVID-19 (Symptoms for > 90 Days) (n = 395)	
Gender								
– female – male – diverse	182 86 0	(67.9%) (32.1%) (0.0%)	67 20 0	(77.0%) (23.0%) (0.0%)	29 13 0	(69.0%) (31.0%) (0.0%)	319 71 2	(81.4%) (18.1%) (0.5%)
Age (mean ± std.dev.)	46.4	± 13.5	41.6	± 11.7	38.1	± 10.8	44.2	± 12.1
Family status								
– single – married – divorced – widowed – separated	97 146 17 4 3	(36.3%) (54.7%) (6.4%) (1.5%) (1.1%)	29 48 8 0 1	(33.7%) (55.8%) (9.3%) (0.0%) (1.2%)	19 22 3 0 0	(43.2%) (50.0%) (6.8%) (0.0%) (0.0%)	128 218 30 5 10	(32.7%) (55.8%) (7.7%) (1.3%) (2.8%)

**Table 2 ijerph-19-09924-t002:** Psychological health status prior to the pandemic.

	Reference Group (n = 274)		Fully Recovered (n = 88)		Prolonged COVID-19 (Symptoms for ≤ 90 Days) (n = 44)		Post-COVID-19 (Symptoms for > 90 Days) (n = 395)	
Depressions (Cramer’s V, *p* > 0.05)	38	(14.4%)	7	(8.3%)	7	(16.3%)	49	(12.6%)
Anxiety disorder (exact test, *p* > 0.05)	21	(8.0%)	6	(7.1%)	1	(2.3%)	27	(6.9%)
Chronic pain (exact test, *p* > 0.05)	31	(11.7%)	5	(6.0%)	2	(4.7%)	52	(13.3%)
Chronic fatigue (exact test, *p* = 0.02)	9	(6.2%)	2	(3.8%)	0	(0.0%)	4	(1.5%)
Burnout (exact test, *p* > 0.05)	12	(4.4%)	1	(1.2%)	3	(7.0%)	10	(2.6%)
Prior psychotherapy (Cramer’s V, *p* > 0.05)	73	(27.7%)	14	(16.7%)	11	(25.6%)	116	(29.7%)
Prior psychiatric treatment (exact test, *p* > 0.05)	15	(5.7%)	1	(1.2%)	3	(7.0%)	34	(8.7%)
Prior psychiatric medication (exact test, *p* > 0.05)	36	(13.6%)	5	(6.0%)	4	(9.3%)	63	(16.5%)

## Data Availability

The datasets generated and analyzed during the current study are not publicly available in accordance with the decision of the involved Research Ethics Boards but are available from the corresponding author on reasonable request within a data-sharing agreement.

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
