# Peer review of "Effects of Pre-Existing Mental Conditions on Fatigue and Psychological Symptoms Post-COVID-19"

_ijerph, 2022, doi:10.3390/ijerph19169924_

Round 1

Reviewer 1 Report

The article is interesting and raises important issues. The way the information is presented is clear. The conclusions presented have scientific value and practical implications. They are important not only for COVID patients, whose numbers are likely to increase with the next waves of the pandemic, but also relevant to mental health prevention in general. I have some doubts and questions for the authors

For Figure 3, The information given in the in the text and in the figure is different. While describing the variable fatigue, the authors wrote about the absence of significant differences between the prolonged and post-Covid group (if I understood correctly - verse 180), but in the figure the difference is marked as significant (.05).

When analysing the variable: anxiety, the figure indicates that there was no difference between the COVID reference and prolonged group - is it true?

In the discussion, the number of participants in the study is given as 893 (verse 196), while in the abstract (verse 6) the number given is 801, which is consistent with the values given in the tables. 

Author Response

Dear Madam or Sir,

Thank you very much for your thorough reading and appreciative feedback. We have addressed your points as follows:

  • Figure 3: Thank you for pointing us to this point. The figure is right as it was generated from the dataset - we've corrected this in the text.
  • Anxiety in Fig.3: We've double-checked the values. Again the figure is correct in that reference and prolonged group do not differ significantly (p = 0.1). This is due to the lower number of persons in the prolonged group.
  • We clarified in discussion that only 801 participants were included in the evaluation.

Reviewer 2 Report

I read this study with great interest, as it presents with an interesting hypothesis and tries to tackle a very important issue: pre-existing mental health conditions of post-COVID-19 patients. I have a couple of remarks:

Although the text is usually unambiguously interpretable, there are minor grammatical mistakes throughout (for example physic disorder instead of physical disorder, or psychologic instead of psychological); therefore a native English editing would be beneficial for clear understanding.

Introduction: A more detailed introduction of previous empirical studies assessing connections of COVID-19 and mental health issues would strengthen the empirical background of the present study.

 Methods: Authors stated that the data collection was separated into two parts due to long fill-out time. How long was the estimated time of filling out the questionnaire? In what part was the interruption?

Results: Were group differences of examined demographic variables between the 4 groups assessed? In case of significant group differences, analyses of mental health variables should be corrected for them.

Figure 3: indication of non-significant result is 1) irrelevant, 2) makes the figure less legible.

Discussion: In light of potential changes of results, it may need revision.

Author Response

Dear Madam or Sir,

Thank you for your detailed review of our manuscript. We hope we could take advantage of your points for the benefit of future readers. Concerning your remarks, we made the following changes:

  • English editing: The paper was corrected by an external corrector, numerous changes were made throughout the text.
  • Introduction: We agree with this point. Therefore the introduction was extended by these two paragraphs: "A recent systematic review by Schou et al. [#schou2021psychiatric] confirms fatigue, anxiety, and depression among the most common sequelae of COVID-19. Anyhow, Magnusdottir et al. [#magnusdottir2022acute] stress that little is known about the long-term mental health of COVID-19 patients. In their observational study across six nations, they find that the severity of the acute disease plays a crucial role: patients with a light course of disease exhibited lower risk of depression and anxiety relative to the reference group – while the opposite was found for patients bedridden for more than seven days.
    Pre-existing mental disorders are reported to increase the risk of COVID-19 severity and mortality [#toubasi2021meta]. There is some agreement that a previous psychiatric history also is a predictor of depression and anxiety [#mazza2021persistent, #romero2021sequelae, #sykes2021post, #toubasi2021meta] following an acute COVID-19 infection. Furthermore, it was hypothesized that pre-existing psychological and psychosomatic conditions are associated with the occurrence and worsening of psychological post-COVID-19 symptoms and fatigue [#mazza2020anxiety, #rudroff2020post]."
  • Methods: The median fill-out time of each surveys was about 30mins and participants were asked to answer second part a few days later. We've added this information.
  • Results: Yes, the description of the group differences was a bit unclear. We rephrased as: "Table [tab1] compares all four groups with respect to gender, age, and family status. No significant differences in age and family status were found. Only the gender variable showed differences between the groups: there were significantly more women (81.4%) in the post-COVID-19 group than in the other three groups (reference 67.9%, recovered 77.0%, prolonged COVID-19 69.0%)."
  • Discussion: We've added a remark that our finding of reduced depression and anxiety in the recovered group accords with the new study by Magnusdottir et al.
  • Figure 3: Thanks you for this hint, this change certainly makes the figure more easy to catch.